# The Unusual Architecture of RNA-Dependent RNA Polymerase (RdRp)’s Catalytic Chamber Provides a Potential Strategy for Combination Therapy against COVID-19

**DOI:** 10.3390/molecules28062806

**Published:** 2023-03-20

**Authors:** Kamel Metwally, Nader E. Abo-Dya, Mohammed Issa Alahmdi, Maha Z. Albalawi, Galal Yahya, Aimen Aljoundi, Elliasu Y. Salifu, Ghazi Elamin, Mahmoud A. A. Ibrahim, Yasien Sayed, Sylvia Fanucchi, Mahmoud E. S. Soliman

**Affiliations:** 1Department of Pharmaceutical Chemistry, Faculty of Pharmacy, University of Tabuk, Tabuk 71491, Saudi Arabia; 2Department of Medicinal Chemistry, Faculty of Pharmacy, Zagazig University, Zagazig 44519, Egypt; 3Department of Pharmaceutical Organic Chemistry, Faculty of Pharmacy, Zagazig University, Zagazig 44519, Egypt; 4Department of Chemistry, Faculty of Science, University of Tabuk, Tabuk 71491, Saudi Arabia; 5Pharm D Program, Faculty of Pharmacy, University of Tabuk, Tabuk 71491, Saudi Arabia; 6Department of Microbiology and Immunology, Faculty of Pharmacy, Zagazig University, Zagazig 44519, Egypt; 7Molecular Bio-Computation and Drug Design Laboratory, School of Health Sciences, University of KwaZulu-Natal, Westville Campus, Durban 4001, South Africa; 8CompChem Lab, Chemistry Department, Faculty of Science, Minia University, Minia 61519, Egypt; 9Protein Structure-Function Research Unit, School of Molecular and Cell Biology, University of the Witwatersrand, Johannesburg 2050, South Africa

**Keywords:** catalytic chamber, RNA-dependent RNA polymerase, combination therapy, COVID-19

## Abstract

The unusual and interesting architecture of the catalytic chamber of the SARS-CoV-2 RNA-dependent RNA polymerase (RdRp) was recently explored using Cryogenic Electron Microscopy (Cryo-EM), which revealed the presence of two distinctive binding cavities within the catalytic chamber. In this report, first, we mapped out and fully characterized the variations between the two binding sites, BS1 and BS2, for significant differences in their amino acid architecture, size, volume, and hydrophobicity. This was followed by investigating the preferential binding of eight antiviral agents to each of the two binding sites, BS1 and BS2, to understand the fundamental factors that govern the preferential binding of each drug to each binding site. Results showed that, in general, hydrophobic drugs, such as remdesivir and sofosbuvir, bind better to both binding sites than relatively less hydrophobic drugs, such as alovudine, molnupiravir, zidovudine, favilavir, and ribavirin. However, suramin, which is a highly hydrophobic drug, unexpectedly showed overall weaker binding affinities in both binding sites when compared to other drugs. This unexpected observation may be attributed to its high binding solvation energy, which disfavors overall binding of suramin in both binding sites. On the other hand, hydrophobic drugs displayed higher binding affinities towards BS1 due to its higher hydrophobic architecture when compared to BS2, while less hydrophobic drugs did not show a significant difference in binding affinities in both binding sites. Analysis of binding energy contributions revealed that the most favorable components are the ΔE_ele_, ΔE_vdw_, and ΔG_gas_, whereas ΔG_sol_ was unfavorable. The ΔE_ele_ and ΔG_gas_ for hydrophobic drugs were enough to balance the unfavorable ΔG_sol_, leaving the ΔE_vdw_ to be the most determining factor of the total binding energy. The information presented in this report will provide guidelines for tailoring SARS-CoV-2 inhibitors with enhanced binding profiles.

## 1. Introduction

Coronaviruses (CoVs) have been associated with significant disease outbreaks in East Asia and the Middle East over the last two decades [1]. In 2002 and 2012, severe acute respiratory syndrome (SARS) and Middle East respiratory syndrome (MERS-CoV) started to emerge. In December 2019, a coronavirus disease outbreak (COVID-19) occurred in Wuhan, Hubei Province, China. The outbreak was initiated by the new virus SARS-CoV-2, which is the seventh member of the CoV family. It quickly spread to nearly every part of China and surrounding countries until it became a worldwide pandemic and global crisis [1,2]. The COVID-19 pandemic is the third severe acute respiratory syndrome coronavirus 2 (SARS-CoV-2), which has led to over two million deaths worldwide [3,4]. Recently, several vaccines have been produced to protect against the novel coronavirus disease (COVID-19); however, those infected with COVID-19 need potent antiviral medications to be cured [5]. Although a few drugs were approved for emergency uses, a promising drug with well-proven clinical efficacy is yet to be discovered. Hence, researchers are continuously attempting to search for potential drug candidates targeting the well-established enzymatic targets of the virus, such as RdRp [5]. 

The structures of many coronavirus proteins have been determined by X-Ray crystallography and Cryogenic Electron Microscopy (Cryo-EM). Since the first outbreak of SARS-CoV-1 in 2002, a high-resolution atomic structure of the coronavirus RNA polymerase has long remained elusive. The first Cryo-EM structure of the SARS-CoV-1 RNA-dependent RNA Polymerase (RdRp) was reported recently in 2019 [3]. Since then, the COVID-19 pandemic has fueled scientific interest in coronavirus biology, and this has led to rapid progress in the structural characterization of coronavirus replication and gene expression. Months after the first case reports, a Cryo-EM structure of the SARS-CoV-2 RdRp was reported, and this was quickly followed by identification of other structures of polymerase complexes in different functional states [4]. Cryo-EM resolved the first structure of the RdRp complex of the novel SARS-CoV-2 virus in April 2020, followed by two other studies that reported similar structures [6,7]. RNA-dependent RNA polymerase (RdRp) is a crucial enzyme in SARS-CoV-2 because it is responsible for genome replication and gene transcription [8]. The RdRp complex is built up from several nonstructural proteins, which are nsp12, nsp7, and nsp8. The protein nsp12 represents the core component and the catalytic subunit of RdRp, while nsp7 and nsp8 are accessory factors that increase the binding and processivity of the RdRp template [7]. The vitality of RdRp in the viral life cycle makes it an excellent target for antiviral drugs, especially for those that have a nucleotide analog scaffold structure, such as remdesivir [8,9,10].

Recently, Cryogenic Electron Microscopy (Cryo-EM) was used to determine the first structure of a small molecule, non-nucleotide analog, suramin, bound to SARS-CoV-2 RdRp [11]. This drug has been found to be effective against both parasite and viral infection. The in vitro studies showed that suramin might also inhibit SARS-CoV-2 replication as well [12]. Interestingly, the structure of RdRp revealed a surprising and unfamiliar feature that has an active catalytic site that encompasses two distinct binding cavities within a so-called “catalytic chamber” (Figure 1).

The structure also shows that two suramin molecules bind to SARS-CoV-2 nsp12 in two different sites with distinctive interaction patterns [11]. This may be due to the differences in amino acid residues and the architecture of each binding pocket. These findings have prompted us to explore this uncommon feature of the so-called “catalytic chamber”.

This project set out to address various questions and aspects; most importantly: (i) how the two binding cavities are conserved/varied in terms of their amino acid sequences and architecture, (ii) the preferential binding landscape of different antiviral drugs in relation to each binding pocket, and (iii) the optimal co-inhibition therapeutics and whether combination therapy would provide additive or synergistic therapeutic effects. In this report, we reveal for the first time the structural and architectural characterization of the two binding pockets of the “catalytic chamber” of the SARS-CoV-2 RdRp at an atomistic level and the chemical and structural aspects that govern the preferential binding of drugs. Through the use of in silico and bioinformatics tools, we analyzed the preferential binding landscape of eight antiviral drugs (remdesivir, suramin, favilavir, ribavirin, molnupiravir, sofosbuvir, alovudine, and zidovudine) (Figure 2) in each of the binding pockets: binding site 1 (BS1) and binding site 2 (BS2). We opted to select these particular antiviral drugs based on the fact that they have been reported to exhibit potential inhibition activities against SARS-CoV-2 RdRp [11,14].

We believe that the information provided in this report will pave the road to a new era of combination therapy protocols against COVID-19.

## 2. Methods and Results

### 2.1. SARS-CoV-2 RdRp Catalytic Chamber

#### Mapping and Characterization of the Two Binding Pockets, BS1 and BS2

Identification of the two binding cavities within the catalytic chamber of SARS-CoV-2 RdRp was carried out based on the experimentally resolved structure of RdRp in complex with two suramin molecules (PDB ID: 7D4F) (Yin et al., 2021) [11]. The protein was stripped of non-standard residues, such as co-factors and ions, excluding the two molecules of suramin bound to the experimentally identified binding sites, BS1 and BS2. The binding site residues that house the suramin molecules, 1 and 2, were determined by mapping out amino acids that lie within 5 Å from each suramin molecule. The identified amino acids sequence of binding of BS1 is: I494, V495, N496, N497, K500, A558, R569, H572, Q573, L576, K577, A580, V588, I589, G590, T591, G683, A685, Y689, L758, and C813. However, the amino acids that constitute the binding site of BS2 are: H439, F480, I548, S549, A550, K551, R553, A554, R555, R836, I837, A840, G852, R858, S861, L862, and D865.

BS1 and BS2 were characterized using physicochemical attributes, such as size, volume, degree of enclosure or exposure, degree of contact, hydrophobic/hydrophilic characteristics, hydrophobic/hydrophilic balance, and hydrogen-bonding possibilities (acceptors/donors), which are presented in Table 1.

## 3. Structural Architecture of BS1 and BS2

Structural architecture and physicochemical attributes of binding sites are critical determinants in the drug design of potential inhibitors. These determinants guide the identification of structural aspects and choice of inhibitors that may exhibit optimal binding affinities [15]. The size of a binding site is determined by the number of site points that make up the binding site. As a rough rule of thumb, two to three site points typically correspond to each atom of the bound ligand, including hydrogens [15]. Our results suggest that BS2 with corresponding size 226 has a larger size compared to BS1 with size 129, as shown in Table 1 and Figure 3B. This interesting finding could also mean that BS2 has the potential to be the preferred binding pocket for relatively larger inhibitors, as opposed to BS1. Nevertheless, our further studies will shed more light on this finding. Subsequently, the volume of a binding site depicts the dimensions of the pocket when considering length, weight, and width as well as the depth of the binding pocket [15]. The basic criteria in determining the average volume for a druggable binding site require that the prospective binding site should have a sitescore > 0.8 and a Dscore > 0.83. BS2 (646.898 A^3^) and exhibit a relatively deeper cavity with a greater volume when compared to BS1 (276.458 A^3^), as shown in Figure 3. We assume that BS2 is larger in size and volume compared to BS1 due to its geometrical location near the entrance of the catalytic chamber where the potential RNA primer strand binds. Exploring the size and volume of the catalytic chamber of SARS-CoV-2 RdRp could reveal vital information that underlies the uncommon mechanism of inhibition in this region necessary to halt viral replication.

## 4. Hydrophobicity/Hydrophilicity Profiles of BS1 and BS2

The hydrophobic and hydrophilic properties of the amino acid residues of the two binding pockets were also compared. These properties, labeled phob and phil in Table 1, measure the relative hydrophobic and hydrophilic attributes of the binding sites. The phobic and philic scores were calibrated so that the average score for a tight-binding site is 1. BS1 shows a slightly lower (0.252) hydrophobicity score compared to BS2 (0.286); however, both binding sites display below-average hydrophobic characteristics, suggesting that only a few of the constituent residues are hydrophobic, as displayed in (red) Figure 4. Likewise, the hydrophilic scores for both binding sites are above average, exceeding the average score of 1. However, BS1 (1.357) is slightly more hydrophilic than BS2 (1.219), as shown in (blue) Figure 4. Conclusively, BS2 has a higher hydrophobic score and hydrophilic score than BS1 and may be more suitable to molecules with such properties. Although the hydrophilic scores of both BS1 and BS2 are above average, they include a few amino acids that are hydrophobic. This pocket architecture allows hydrophobic amino acids to move to the core of the protein to avoid water, and the hydrophilic side chains move towards the outside of the protein to be more exposed to surrounding water. Together, these observations may be considered in the design of inhibitors with preferential binding profiles towards one pocket over the other.

## 5. BS1 and BS2 Binding Pocket Per-Residue Contribution Using Suramin as a Prototype

Per-residue energy contribution of BS1 and BS2 amino acids towards the binding of each suramin molecule was computed using MolDock scoring function [16] to elaborate on the binding landscape of each site and whether they are conserved or if they varied significantly in terms of their binding themes. Our per-residue energy contribution analysis showed that the two binding sites contribute differently towards the overall binding of each suramin molecule (Table 2). The active site residues (Asn496, Ile494, Arg569, Lys577, Gly590, and Lys500) of BS1 were found to show the highest contribution towards the overall binding to suramin 1; however, the residues (Arg555, Arg536, Arg553, LIe548, Ser549, and Lys551) in BS2 are the major contributors to suramin 2 binding. These variations in energy contributions between residues of the two distinct binding sites could define their uniqueness towards inhibitor binding.

## 6. System Preparation and Molecular Dynamic Simulation

The X-ray crystal structures of RNA-dependent RNA polymerase (RdRp) were retrieved from the Protein Data Bank [17] (PDB code: 7D4F) [11]. These structures were then prepared for molecular dynamic (MD) simulation using the UCSF Chimera software package [18]. MarvinSketch 6.2.1, 2014, and Molegro Molecular Viewer (MMV) were used for the ligands preparation and ensured that the ligands’ proper angles and hybridization state were displayed [16,19]. AutoDock Tools GUI was used to describe the grid box at the catalytic site of the protein [20]. The dimensions and co-ordinates of the grid box for binding site 1 (BS1) were defined as follows: size x = 9.78025, y = 16.3064, z = 9.20736 with center x = 119.265, y = 133.527, z = 145.034 and size x = 19.0436, y = 11.5639, z = 10.3082 with center x = 129.422, y = 124.624, z = 150.334 for binding site 2 (BS2). The Lamarckian genetic algorithm was used to perform docking calculations [21]. The prepared systems protonation states were optimized using Maestro Schrödinger [13]. The necessary hydrogen atoms were corrected, and the neutral residues were capped to ensure protein stability during the simulation. AutoDock Vina’s highest scoring docked pose was used as the initial structure for a molecular dynamics (MD) simulation run [22,23].

Altogether, sixteen systems comprising the binding of the eight antivirals with each binding site of RdRp enzyme were subjected to MD simulations using the Graphic Process Unit version of the AMBER18 software package [24]. Protein optimization and explicit solvation were carried out using the integrated LEAP module, while the AMBER18 software forcefield was employed to define protein parameters [25]. The systems were partially minimized for 2500 steps with a restraint potential of 500 kcal/mol Å, followed by full minimization of 10,000 steps. The systems were gradually heated from 0–300 K using a Langevin thermostat in a canonical ensemble (NVT) [26]. Equilibration was also carried out without restraints at a temperature of 300 k in an NPT ensemble for 1000 ps while atmospheric pressure was maintained at 1 bar using the Berendsen barostat [27]. This was followed by MD production runs of 100 ns for each system, during which the SHAKE algorithm was used to constrict all atomic hydrogen bonds [28]. The integrated CPPTRAJ and PTRAJ modules [29] of AMBER18 were used to analyze resulting coordinates and trajectories while obtained data were plotted using Microcal Origin software (www.originlab.com) [30]. UCSF Chimera was also used to visualize and analyze structural events. These are in accordance with our in-house MD simulation protocol, which has been previously reported [31,32].

## 7. Dynamic Conformational Stability and Fluctuations

MD simulations were carried out to investigate the inhibition performance and interactions of the potential eight ligands with BS1 and BS2. Validation of system stability and flexibility is essential for tracing disrupted motions and avoiding artifacts that may arise during the course of the simulation [33,34]. In this study, root-mean-square deviation (RMSD) and root-mean-square fluctuation (RMSF) were calculated to measure the systems’ stability and flexibility during the 100 ns simulations [35,36]. The tracing to disrupt movements and prevent artifacts that could appear during the simulation needs the stability of a system to be validated. Therefore, we evaluated the stability of eight inhibitors inside the BS1 and BS2. The orientation that the ligand displays within a specific binding site may have an impact on ligand stability, as the therapeutic impact of a small molecule depends on its stability in a target protein’s binding region. The root-mean-square deviation measures the difference between a protein’s backbones from its initial structural conformation to its final position. However, the residual conformational analysis is a measure of the nature of fluctuation exhibited by individual residue corresponding to the effect of ligand induction on the protein, cumulatively yielding its therapeutic efficacy. For all systems, RMSF was calculated for each amino acid residue during MD simulation of 100 ns. Furthermore, to indicate how the protein surface interrelates with solvent atoms and how it relays to the compactness of the hydrophobic protein core, the solvent-accessible surface area (SASA) of the protein upon ligand binding was calculated. This was accomplished by computing the surface area of the protein observable to solvent across the 100 ns MD simulation, which is vital for biomolecular stability. According to this analysis, the average values of RMSD, RMSF, and SASA of the eight inhibitors within BS1 and BS2 are presented in Table 3 and Table 4. Additionally, a structural visualization using simulation RMSD, RMSF, and SASA post-analyses for the eight inhibitors inside the BS1 and BS2 are shown in Appendix A, respectively.

## 8. Binding Free Energy Calculations

In order to estimate the binding interactions of these antiviral drugs to RdRp enzyme, binding free energy calculations were carried out using the molecular mechanics/Poisson–Boltzmann surface area (MM/PB-SA) method [37,38]. This approach is widely employed and proven to be reliable in measuring binding free energies involved in protein–ligand complex formation. Moreover, MM/PBSA is mathematically represented as follows:ΔG_bind_ = G_complex_ − G_receptor_ − G_ligand_(1)
E_gas_ = E_int_ + E_vdw_ +E_ele_(2)
G_sol_ = G_GB/PB_ + G_SA_(3)
G_SA_ = γSASA(4)
where van der Waals and electrostatic interactions are represented as E_vdw_ and E_ele_ while E_gas_ denotes gas-phase energy and E_int_ as internal energy. The solvation free energy denoted by G_sol_ represents the solvation free energy and can be decomposed into polar and nonpolar contribution states. The polar solvation contribution, G_GB/PB_, is determined by solving the GB/PB equation, whereas G_SA_, the nonpolar solvation contribution, is estimated from the solvent-accessible surface area (SASA), determined using a water probe radius of 1.4 Å. Per-residue decomposition analyses were also carried out to estimate individual energy contribution of binding site residues to the stabilization and affinity of remdesivir, suramin, favilavir, ribavirin, molnupiravir, sofosbuvir, alovudine, and zidovudine. This could provide more insights into the basis of the RdRp inhibition exhibited by these drugs as high residual energy contributions could depict crucial residues.

## 9. Assessment of Comparative Binding Energies

An assessment of the comparative binding energies for RdRp in complex with the eight antiviral drugs at the two distinct binding sites was conducted for the 100 ns simulation run. The molecular mechanics/generalized Born surface area (MM/GBSA) method for predicting binding energies was employed to estimate the binding affinities of each of the eight antiviral drugs (Figure 2) at the two binding pockets of the RdRp enzyme (Table 5).

The MM/GBSA and molecular mechanics Poisson−Boltzmann surface area (MM/PBSA) are very popular methods for binding energy prediction and are known to be more accurate than most scoring functions in molecular docking. They are also computationally less demanding than alchemical free energy methods. As shown in Table 3, this study revealed that all antiviral drugs bind favorably to the two binding pockets due the high ΔG_bind_ values estimated for all systems. Furthermore, the energy terms contribute to the binding free energy, with the most favourable components being ΔE_ele_, ΔE_vdw_, and ΔG_gas_, while ΔG_sol_ is unfavorable. The ΔE_ele_ and ΔG_gas_ are enough to balance the unfavorable ΔG_sol_, leaving the ΔE_vdw_ as the determining factor of the total energy.

In BS1, remdesivir demonstrates the most favorable binding, with an estimated binding energy of −52.56 kcal/mol, while favilavir is the least favourable, with an estimated binding energy of −11.38 kcal/mol. Similarly, in the BS2 system, remdesivir shows the strongest binding among all the antiviral drugs, with an energy of −24.55 kcal/mol, while alovudine shows the least binding, with an estimated energy of −10.38 kcal/mol. A comparison of energies of all antiviral drugs across the two binding sites reveals that five out of the eight antiviral drugs show stronger binding in BS1 compared to their energies in BS2 comprising remdesivir (from −52.56 kcal/mol to −24.55 kcal/mol), sofosbuvir (from −24.34 kcal/mol to −17.75 kcal/mol), alovudine (−16.99 kcal/mol to −10.38 kcal/mol), zidovudine (from−16.82 kcal/mol to −11.12 kcal/mol), and suramin (−13.13 kcal/mol to −12.70 kcal/mol). These observed drops in energies among all five drugs in BS2 could suggest that BS1 is the most preferred binding site for remdesivir, sofosbuvir, alovudine, zidovudine, and suramin. Likewise, in BS2, molnupiravir, favilavir, and ribavirin show stronger energies as compared to their energies in BS1, suggesting that BS2 could be the most preferred binding site for these drugs. Conclusively, based on these findings, the preferred binding site is dependent on the specific inhibitor involved, as the inhibitors used herein demonstrate affinity to either of the binding pockets.

## 10. Conclusions

The interesting architecture of the catalytic chamber of RNA-dependent RNA polymerase (RdRp) with its two unique binding sites, BS1 and BS2, prompted us to investigate the potential use of multiple inhibitors as combination therapy against COVID-19. The architecture of the binding sites BS1 and BS2 within the catalytic chamber and the preferential binding modes of eight antiviral drugs in each binding site were thoroughly analysed in order to understand the underlying factors that govern preferential drug binding. Analysis of data obtained in this study revealed that the hydrophobic drugs remdesivir and sofosbuvir were found to bind better in both binding sites than the relatively less hydrophobic drugs alovudine, molnupiravir, zidovudine, favilavir, and ribavirin, with the exception of suramin, which displayed overall weaker binding affinities in both binding sites when compared to other drugs, despite its apparently high hydrophobic character. The low binding affinity of suramin may be attributed to its high binding solvation energy, which disfavors its overall binding. Moreover, hydrophobic drugs showed higher binding affinities towards BS1 when compared to BS2, while less hydrophobic drugs did not show a significant difference in binding affinities in both binding sites. This may be attributed to the fact that BS1 has more hydrophobic architecture compared to BS2. Furthermore, our findings revealed the energy terms that contribute to the binding free energy, with the most favourable components being the ΔE_ele_, ΔE_vdw_, and ΔG_gas_, while ΔG_sol_ was unfavorable. The ΔE_ele_ and ΔG_gas_ for hydrophobic drugs were enough to balance the unfavorable ΔG_sol_, making the ΔE_vdw_ the major determining factor of the total energy. The information presented in this report will lay the foundation for further research and development of co-inhibition therapies and represents a major contribution to the field of SARS-CoV-2 research.

## Figures and Tables

**Figure 1 molecules-28-02806-f001:**
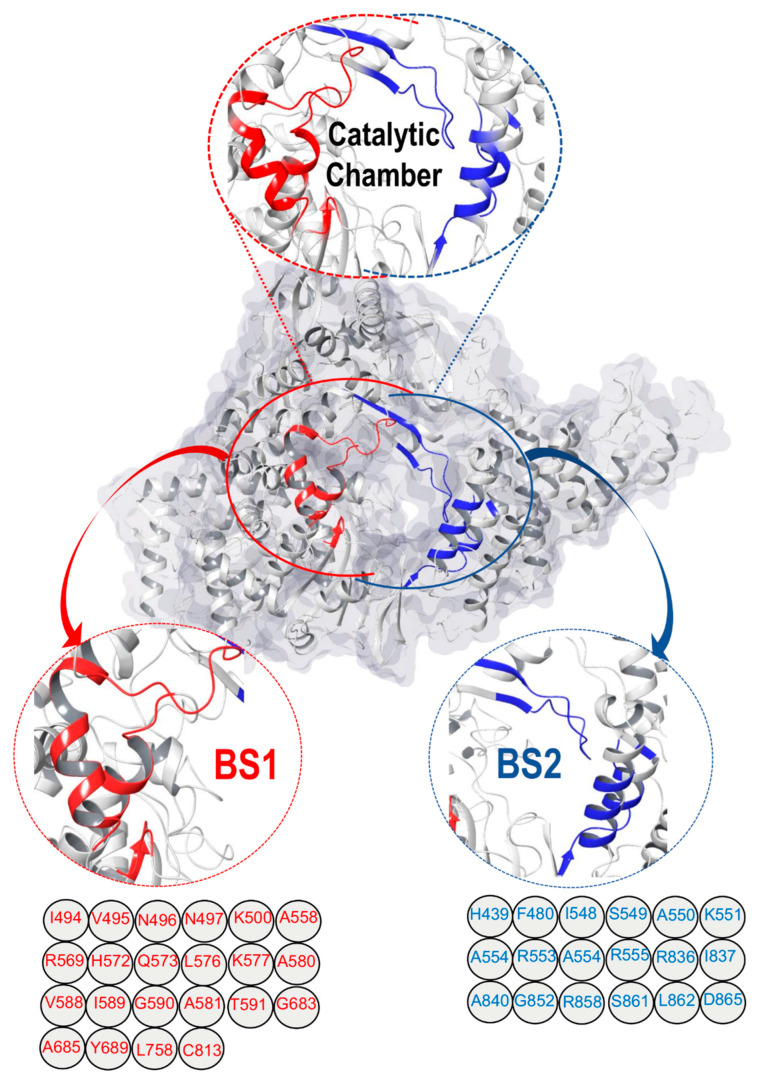
Surface view of RNA-dependent RNA Polymerase (RdRp) (PDB ID: 7D4F) [11], (shows the “catalytic chamber” housing two distinct binding sites/cavities, estimated with the Maestro Schrödinger [13]. The amino acid residues of binding site1 (BS1) and binding site2 (BS2) are also shown.

**Figure 2 molecules-28-02806-f002:**
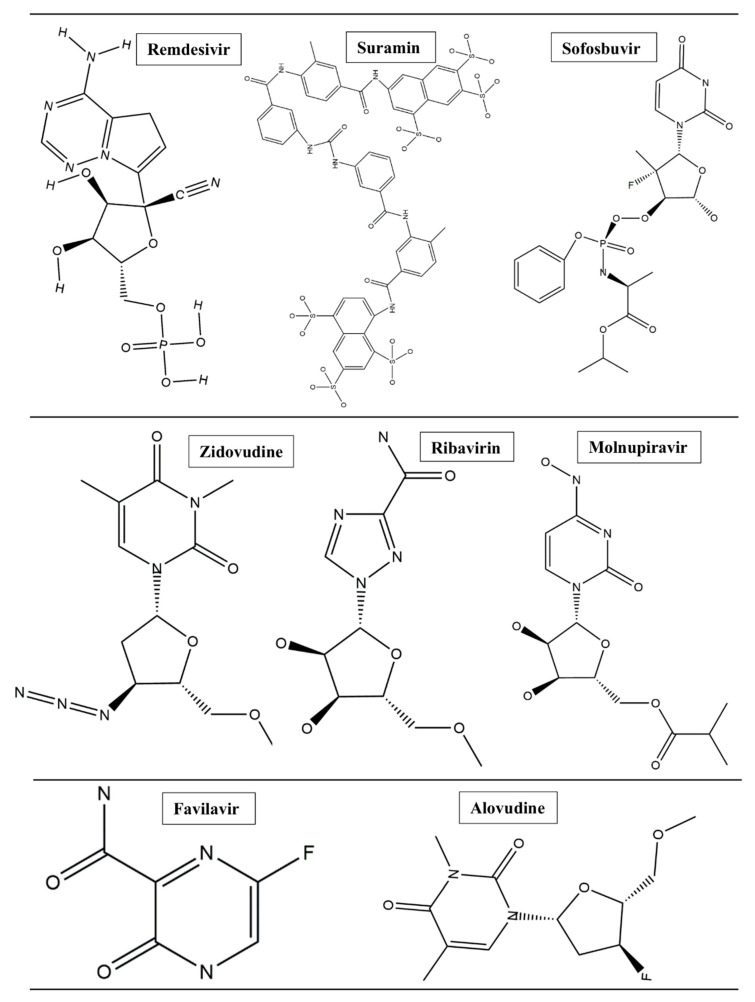
List of the two-dimensional structure of the eight antiviral drugs used for this study.

**Figure 3 molecules-28-02806-f003:**
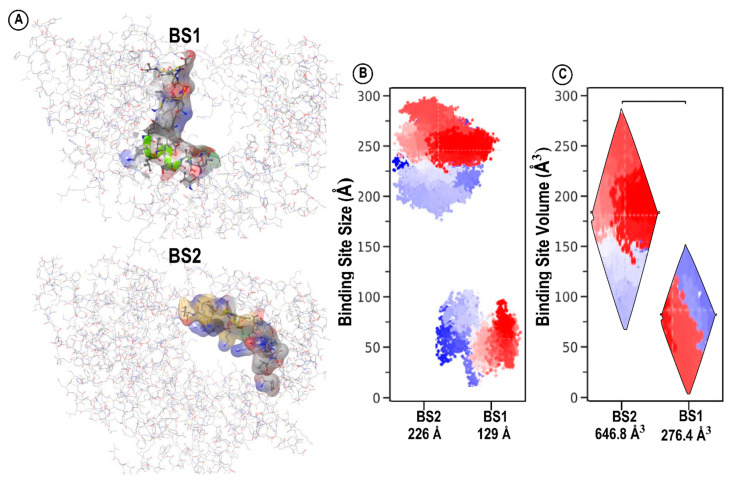
Binding site size and volume analysis. (**A**) BS1 and BS2 of the catalytic chamber amino acid molecular number surface view. (**B**) Binding pockets size comparison of BS1 and BS2. Difference of red and blue motion of both pocket sites. (**C**) The volume of the specificity conferring moiety of the pocket sites, estimated [13], differs significantly between BS1 and BS2.

**Figure 4 molecules-28-02806-f004:**
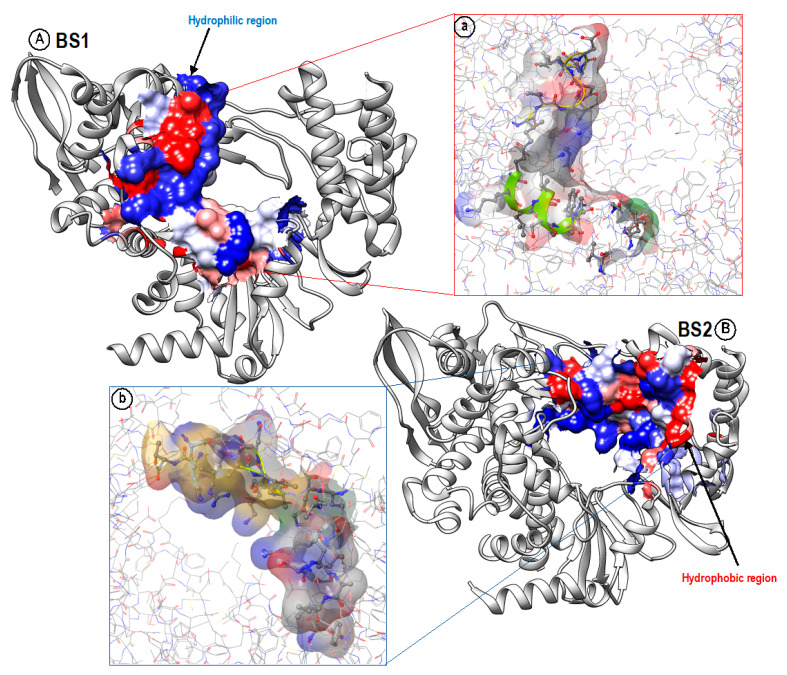
Analyzing the hydrophobic (red) and hydrophilic (blue) regions of (**A**), (**a**) BS1 and (**B**), (**b**) BS2. Both binding pockets have large regions that are hydrophilic, although there are a few hydrophobic regions.

**Table 1 molecules-28-02806-t001:** Characterization of RdRp Binding Sites.

Site	Site Score	Size	Volume A^3^	Dscore	Exposure	Enclosure	Contact	Phobic	Philic	Balance	Don/acc
BS1	1.030	129	276.458	0.943	0.522	0.743	1.027	0.252	1.357	0.186	0.370
BS2	0.976	226	646.898	0.940	0.695	0.662	0.771	0.286	1.219	0.235	0.866

**Table 2 molecules-28-02806-t002:** BS1 and BS2 Per-Residue Energy Contribution Analysis.

BS1/Suramin Complex		BS2/Suramin Complex	
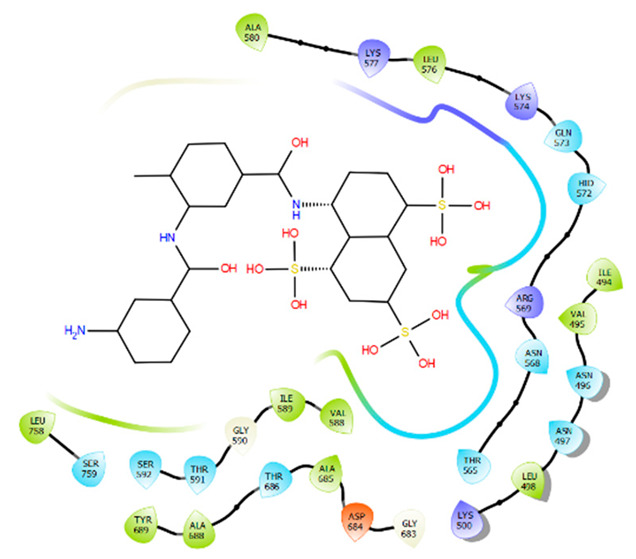		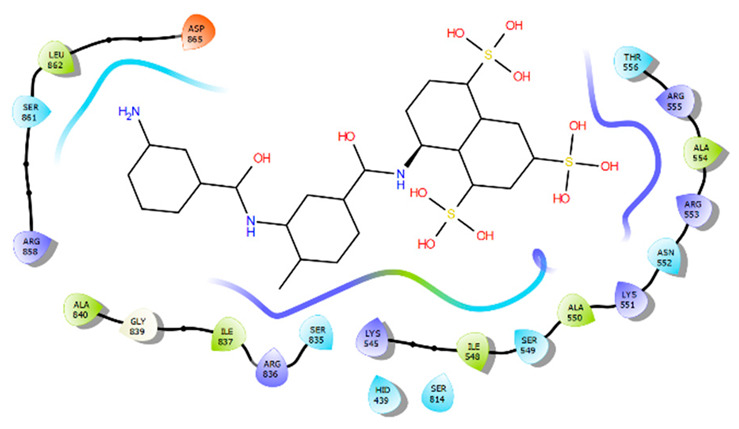	
BS1 per-residue energy contribution		BS2 per-residue energy contribution	
Residue	Energy (kcal/mol)	Residue	Energy (kcal/mol)
Asn496	−22.51	Arg555	−30.26
Ile494	−22.51	Arg836	−19.47
Arg569	−19.44	Arg553	−13.50
Lys577	−16.14	LIe548	−11.90
Gly590	−15.77	Ser549	−11.19
Lys500	−14.60	Lys551	−11.01
Gln573	−8.99	His439	−10.08
Ala558	−7.89	Gly852	−8.93
Ile589	−7.01	Arg858	−5.84
Leu576	−6.80	Ala550	−5.81
Asn497	−6.48	Ala840	−5.48
Ala685	−4.61	Phe480	−5.18

**Table 3 molecules-28-02806-t003:** RMSD, RMSF, and SASA profile of the eight ligands when bound to BS1.

Systems	Estimated Averages (Å)		
Ligand	RMSD	RMSF	SASA
Remdesivir-BS1	1.58	1.06	14,011.99
Sofosbuvir-BS1	2.04	1.12	14,510.63
Alovudine-BS1	1.34	1.03	13,772.77
Molnupiravir-BS1	1.65	1.09	13,800.19
Zidovudine-BS1	1.72	1.08	13,706.09
Favilavir-BS1	1.57	1.17	14,271.97
Ribavirin-BS1	1.53	1.04	14,116.37
Suramin-BS1	1.62	1.20	13,911.23

**Table 4 molecules-28-02806-t004:** RMSD, RMSF, and SASA profile of the eight ligands when bound to BS2.

Systems	Estimated Averages (Å)		
Ligand	RMSD	RMSF	SASA
Remdesivir-BS2	1.87	1.14	11,018.38
Sofosbuvir-BS2	2.09	1.35	11,193.50
Alovudine-BS2	1.80	1.06	10,655.36
Molnupiravir-BS2	2.13	1.21	11,180.05
Zidovudine-BS2	2.12	1.22	11,245.66
Favilavir-BS2	1.64	1.20	10,948.94
Ribavirin- BS2	1.68	1.11	11,244.42
Suramin-BS2	1.78	1.24	11,142.55

**Table 5 molecules-28-02806-t005:** MM/GBSA-based binding free energy profile of each of the eight antiviral drugs at the two binding pockets of RdRp enzyme.

Systems	Energy Components				
	(kcal/mol)				
**Ligand**	ΔE_vdw_	ΔE_ele_	ΔG_gas_	ΔG_sol_	ΔG_bind_
**Remdesivir-BS1**	−41.9522	−335.8092	−377.7613	325.1969	−52.5645
**Remdesivir-BS2**	−42.7075	−90.9436	−133.6512	109.1042	−24.5469
**Sofosbuvir-BS1**	−34.6272	−15.0141	−49.6413	25.2979	−24.3434
**Sofosbuvir-BS2**	−32.0037	−19.3497	−51.3534	33.6064	−17.7470
**Alovudine-BS1**	−21.7154	−8.8406	−30.5559	13.5672	−16.9888
**Alovudine-BS2**	−19.2905	3.337	−15.9536	5.5781	−10.3754
**Molnupiravir-BS1**	−22.4816	−34.6077	−57.0892	43.4611	−13.6282
**Molnupiravir-BS2**	−26.1253	−24.0320	−56.1573	35.6699	−14.4874
**Zidovudine** **-BS1**	−24.3850	−31.0102	−57.3952	40.5715	−14.8237
**Zidovudine** **-BS2**	−15.3793	−193.9331	−209.3124	198.1934	−11.1190
**Favilavir-BS1**	−7.5696	−37.4859	−79.4035	90.7847	−11.3812
**Favilavir-BS2**	−7.3014	−28.6874	−72.1635	85.046	−12.8825
**Ribavirin-BS1**	−8.5964	−20.8937	−63.3037	76.6451	−13.3414
**Ribavirin-BS2**	−7.7187	−27.1752	−69.8784	83.4503	−13.5719
**Suramin-BS1**	−8.6775	−48.0019	−89.7680	102.9012	−13.1331
**Suramin-BS2**	−7.7503	−51.9245	−94.8926	107.5886	−12.6959
**All energies are in kcal/mol.**	ΔE_ele_ = electrostatic energy	ΔE_vdw_ = van der Waals energy	ΔG_bind_ = total binding free energy	ΔG_sol_ = solvation free energy	ΔG = gas phase free energy

## Data Availability

All data are available for future work.

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
