# Peer review of "The Unusual Architecture of RNA-Dependent RNA Polymerase (RdRp)’s Catalytic Chamber Provides a Potential Strategy for Combination Therapy against COVID-19"

_molecules, 2023, doi:10.3390/molecules28062806_

Round 1

Reviewer 1 Report

The manuscript entitled "A Key to a New Era of Combination Therapy Against COVID-19 by Exploiting the Unusual Architecture of RNA-Dependent RNA Polymerase (RdRp)'s Catalytic Chamber and Preferential Antiviral Drug Binding" by Kamel Metwally et al. for publication in this journal. I have gone through the manuscript and the work done by the authors is new, but major changes are required in order to get accepted in this reputed journal.

-The title of the manuscript should be improved.

-In the introduction section, the author needs to discuss some more recent reports. The information provided is not sufficient. Some recent articles need to be cited too; 10.1039/D1ME00147G; 10.1007/s11224-022-01996-y; etc.

-The aim of the paper should be a little more highlighted in the last paragraph of the introduction.

-Please check the entire manuscript for grammatical errors.

-Methodology section should be distinguished and more details should be provided. Molecular docking methods do not contain important details, e.g., the size of the grid, number of grid points, and the center of the box are also missing. For molecular docking studies, go through the following recent articles and cite time accordingly- DOI: 10.3390/ph15030285; 10.3390/molecules27051724

-In Table 2, figure overlaps the text.

-MD Simulation part needs to be improved. What about RMSD, RMSF, Rg, SASA of the complexes? How many ions are used in the neutralization step? Refer to the recent articles about MD simulation, improve this section, and cite them to make it more impressive- DOI: 10.3390/molecules27103295; 10.3390/molecules27103290

-Conclusion should be rewritten in a more informative way.

-More efforts should be made to explain the application of the current work.

-Novelty should be discussed.

-Ab initio quantum chemistry methods could be used for calculations.

Author Response

Kindly, find the answer for the comments in the table attached.

All the best.

Reviewer 2 Report

In the manuscript by Metwally et al., the authors analyzed the binding features of  SARS-Covid-2 RdRp catalytic chamber with 8 antiviral drugs in silico. Based on the analyses, binding site 1 (BS1) in the catalytic chamber is more hydrophobic and has higher binding affinities with hydrophobic drugs. Comparing different binding features, ΔGsol is unfavorable but balanced by Δ???? and Δ?gas, and Δ???? may be the major determining factor of the total binding energy. 

Major issues:

- The authors compared 8 antiviral drugs. However, they did not introduce of all drugs with references, and did not give reasons to pick those drugs for comparison.

- The terms are not unified in the manuscript, for example, ΔGsol is bold, while ΔGgas is italic. Please check them.

- Have the binding affinities (in table 3) calculated in silico been tested by experiments (in literature)? The authors calculated the affinities with BS1 and BS2, how about the affinities of the entire catalytic chamber with drugs? How can those features guide to select drugs for testing, could you discuss more?

Minor issues:

- Line 196, "Lie548" should be "Ile548". Same as in the table 2.

- Table 2, the pictures covered other rows. Why the structure of suramin here is different from figure 4?

- Line 258, the backslash should be removed.

- Table 3 The numbers are not aligned and data of Remdesivir-BS1 were missingIn the bottom row, "ele" in "Δ?ele" should be subscript. Other terms also need to be checked.

Author Response

kindly, find the answers to the comments in the tables attached.

All the best.

Round 2

Reviewer 2 Report

Line 190, "LIe548" should be "Ile548". Same as in the table 2. Please check them again!